# Cardiac Myosin Inhibitors in Hypertrophic Cardiomyopathy: From Sarcomere to Clinic

**DOI:** 10.3390/ijms26199347

**Published:** 2025-09-24

**Authors:** Kazufumi Nakamura, Takahiro Okumura, Seiya Kato, Kenji Onoue, Toru Kubo, Hidemichi Kouzu, Toshiyuki Yano, Takayuki Inomata

**Affiliations:** 1Department of Cardiovascular Medicine, Okayama University Graduate School of Medicine, Dentistry and Pharmaceutical Sciences, Okayama 700-8558, Japan; 2Center for Advanced Heart Failure, Okayama University Hospital, Okayama 700-8558, Japan; 3Department of Advanced Cardiovascular Therapeutics, Nagoya University Graduate School of Medicine, Nagoya 466-8550, Japan; takaoku@med.nagoya-u.ac.jp; 4Department of Cardiology, Nagoya University Graduate School of Medicine, Nagoya 466-8550, Japan; 5Division of Pathology, Saiseikai Fukuoka General Hospital, Fukuoka 810-0001, Japan; seikato1014@gmail.com; 6Department of Cardiovascular Medicine, Nara Medical University, Kashihara 634-8521, Japan; konoue@naramed-u.ac.jp; 7Department of Cardiology and Geriatrics, Kochi Medical School, Kochi University, Kochi 783-8505, Japan; jm-kubotoru@kochi-u.ac.jp; 8Department of Cardiovascular, Renal and Metabolic Medicine, Sapporo Medical University School of Medicine, Sapporo 060-8543, Japan; kouzu@sapmed.ac.jp (H.K.); tyano@sapmed.ac.jp (T.Y.); 9Department of Cardiovascular Medicine, Niigata University Graduate School of Medical and Dental Sciences, Niigata 951-8510, Japan; inotaka@med.niigata-u.ac.jp

**Keywords:** hypertrophic cardiomyopathy, myosin inhibitors, sarcomere, mavacamten, aficamten, heart failure

## Abstract

Hypertrophic cardiomyopathy (HCM) is a primary myocardial disease characterized by unexplained left ventricular hypertrophy, often resulting from pathogenic variants of sarcomeric protein genes. Conventional treatments, such as the use of beta blockers or calcium channel blockers, focus on symptomatic control but do not address the underlying hypercontractility at the sarcomere level. Recent advances in molecular understanding have led to the development of cardiac myosin inhibitors that directly modulate sarcomeric function by reducing myosin–actin cross-bridge formation and adenosine triphosphatase (ATPase) activity. Mavacamten and aficamten have shown promising results in phase 2 and 3 clinical trials, improving symptoms, exercise capacity, and left ventricular outflow tract gradients in patients with obstructive HCM. This review summarizes the current understanding of HCM pathophysiology, diagnostic strategies, and conventional treatments with a focus on the mechanisms of action of myosin inhibitors, clinical evidence supporting their use, and future directions for improvement. We also discuss their potential applications in non-obstructive HCM and the importance of precision medicine guided by genetic profiling.

## 1. Introduction: What Is Hypertrophic Cardiomyopathy (HCM)?

Hypertrophic cardiomyopathy (HCM) is a primary myocardial disease characterized by unexplained left ventricular hypertrophy, which is often asymmetric, in the absence of secondary causes such as hypertension or aortic stenosis. First described pathologically by Teare in 1958 in young individuals who died suddenly [1], HCM has since been recognized as one of the most common inherited cardiovascular disorders. Genetically, HCM is predominantly inherited in an autosomal dominant manner and is most frequently caused by mutations in sarcomeric proteins, particularly myosin heavy chain 7 (*MYH7*) and myosin binding protein C 3 (*MYBPC3*) [2]. These mutations lead to abnormalities in the structure and function of the cardiac myocytes, resulting in myocyte disarray, fibrosis, and hypercontractility. The phenotypic expression of HCM is highly variable, resulting in a phenotypic range spanning asymptomatic individuals to those with severe heart failure or sudden cardiac death.

Despite the availability of symptom-relieving medications, such as beta blockers and calcium channel blockers, no disease-specific therapy has been developed until recently. A growing understanding of the molecular pathophysiology of HCM, particularly the roles of sarcomeric hypercontractility and increased adenosine triphosphate (ATP) consumption, has paved the way for the development of myosin inhibitors that directly target the underlying cause of the disease [3,4,5].

This review aims to provide a comprehensive overview of the molecular pathology of HCM, rationale for targeting cardiac myosin, and clinical development and future potential of myosin inhibitors as a novel therapeutic strategy.

## 2. Epidemiology, Diagnosis, and Treatment of HCM

### 2.1. Epidemiology

HCM is one of the most common inherited cardiomyopathies, affecting approximately 1 in 500 individuals worldwide [6]. Recent epidemiological analyses have demonstrated an even higher prevalence when undiagnosed and subclinical cases were included [7]. In Japan, a nationwide administrative claims database study estimated a standardized prevalence of 11.1 per 10,000 people in 2021, corresponding to approximately 140,000 patients [8]. Although HCM can occur in all age groups, it is typically diagnosed in adulthood, with a slight male predominance [9,10]. Sudden cardiac death, often secondary to malignant arrhythmia, is the leading cause of mortality, followed by congestive heart failure and cerebrovascular events [10,11].

Phenotypic classification plays a pivotal role in clinical management. Based on morphology, HCM is typically categorized into obstructive (hypertrophic obstructive cardiomyopathy, HOCM), non-obstructive, mid-ventricular obstructive, apical, and dilated-phase or end-stage HCM (Figure 1) [12].

HOCM is defined by a left ventricular outflow tract (LVOT) pressure gradient ≥ 30 mmHg at rest or during provocation and is associated with a higher risk of sudden cardiac death. While HOCM is identified in approximately 35% of patients in Western registries, its prevalence may be higher when including cases of latent obstruction revealed under stress [13]. Apical HCM, characterized by localized hypertrophy of the left ventricular apex, is observed more frequently in Japan, accounting for 15–25% of HCM cases [14,15]. Although often associated with a relatively benign course, complications such as apical aneurysm formation and embolic events can occur.

Traditionally, HCM has been described as a monogenic disorder caused by autosomal dominant mutations in the sarcomere protein genes [2,12]. However, genetic penetrance and expression are highly variable [2,16]. Recent studies have highlighted a complex genetic architecture involving modifier genes, polygenic risk scores, and epigenetic factors that may influence disease expression [17]. Moreover, in patients with atypical features, it is essential to consider phenocopies such as cardiac amyloidosis and Fabry disease [2,12].

### 2.2. Diagnosis

The diagnosis of HCM is based on imaging evidence of unexplained left ventricular hypertrophy in the absence of abnormal loading conditions such as hypertension or aortic stenosis. In adults, HCM is generally defined by a maximal wall thickness of ≥15 mm, whereas a threshold of ≥13 mm may be used for first-degree relatives of affected individuals [12]. Transthoracic echocardiography is the first-line imaging modality that allows the assessment of wall thickness distribution, systolic function, and presence of left ventricular outflow tract obstruction (LVOTO). In cases of suspected HOCM without resting obstruction, provocation maneuvers, such as the Valsalva maneuver, or exercise testing are essential [18].

Cardiac magnetic resonance imaging (MRI) provides superior spatial resolution and tissue characterization. It is particularly useful for identifying apical hypertrophy, quantifying left ventricular volume, and detecting myocardial fibrosis through late gadolinium enhancement. The presence and extent of late gadolinium enhancement are associated with an increased risk of sudden cardiac death as well as progressive heart failure [19]. International guidelines recommend cardiac MRI for patients with inconclusive echocardiographic findings and prognostic assessments [12,20].

Electrocardiography frequently reveals abnormalities, such as increased QRS voltages, ST-T wave changes, and deep negative T waves, particularly in apical HCM. Ambulatory Holter monitoring and exercise testing are crucial for arrhythmic risk stratification, especially in detecting nonsustained ventricular tachycardia, a key predictor in sudden cardiac death risk models. These models incorporate parameters such as age, maximal wall thickness, family history, extent of late gadolinium enhancement, and arrhythmias, and are used to guide implantable cardioverter-defibrillator decisions [21].

Cardiac enzymes such as creatine kinase (CK) or lactate dehydrogenase (LDH) may show mild or transient elevation in some patients with HCM due to ongoing myocyte injury or microvascular ischemia; however, they are not specific for the disease and are therefore not routinely used for its diagnosis. In contrast, circulating natriuretic peptides (brain natriuretic peptide [BNP] or N-terminal pro-brain natriuretic peptide [NT-proBNP]) are often persistently elevated and more closely reflect diastolic dysfunction and hemodynamic burden in HCM, thus providing adjunctive value in disease assessment rather than initial diagnosis.

Although not essential for diagnosis, genetic testing supports family screening and helps distinguish HCM from other phenocopies [22]. The presence of pathogenic sarcomere gene variants supports this diagnosis; however, their absence does not exclude the possibility of HCM.

### 2.3. Treatment

Therapeutic strategies for HCM are individualized based on the phenotype, symptom burden, and comorbid risk. A comprehensive overview of HCM phenotypes, their associated clinical outcomes, and treatment options is presented in Figure 2. Asymptomatic patients with preserved systolic function are generally managed conservatively with regular follow-ups and lifestyle counseling, including avoidance of dehydration and extreme exertion. In contrast, symptomatic patients, particularly those with obstructive physiology, require more proactive interventions due to the direct contribution of dynamic LVOTO to exertional dyspnea, chest pain, and syncope [12,20].

Beta blockers are the first-line pharmacological agents for HOCM treatment. They mitigate LVOT gradients and improve symptoms by reducing the heart rate and myocardial contractility [23]. However, caution is warranted as HCM is characterized by small ventricular cavities and dependence on adequate stroke volume. Excessive beta-blockade may lead to reduced cardiac output and worsening of symptoms. Thus, careful titration is required, which fundamentally differs from the maximally tolerated dose strategy used for treating heart failure with a reduced ejection fraction. If beta blockers are not tolerated or are insufficient, non-dihydropyridine calcium channel blockers (e.g., verapamil) may be employed. Disopyramide, a sodium channel blocker with negative inotropic effects, is often added in refractory cases but requires careful monitoring owing to potential QT prolongation and anticholinergic side effects [24]. In Japan, cibenzoline is also widely used as a negative inotropic agent with sodium channel-blocking properties, and its use has been documented in several clinical reports [25]. In patients with obstructive physiology, vasodilators such as angiotensin-converting enzyme (ACE) inhibitors, angiotensin receptor blockers (ARBs), or angiotensin receptor–neprilysin inhibitors (ARNI, e.g., sacubitril/valsartan) may exacerbate LVOTO and are therefore generally avoided [12].

In patients with drug-refractory symptoms and significant obstruction (resting or provoked LVOT gradient ≥ 50 mmHg), invasive septal reduction therapy should be considered. Surgical septal myectomy remains the gold standard, especially in younger patients or those with concomitant mitral valve abnormalities or other cardiac diseases. Alcohol septal ablation offers a catheter-based alternative in carefully selected patients [26,27]. When performed at experienced centers, both procedures substantially improve the symptoms and quality of life.

Importantly, the fundamental pathophysiology of HCM involves hypercontractility due to excessive actin–myosin cross-bridge cycling, which underlies not only LVOTO but also diastolic dysfunction and myocardial fibrosis [28]. This insight has led to the development of cardiac myosin inhibitors (CMIs), a novel class of therapeutics that directly target sarcomere overactivity. By modulating myosin adenosine triphosphatase (ATPase) activity and reducing the number of force-generating cross-bridges, agents such as mavacamten and aficamten decrease hypercontractility without compromising systolic function, resulting in reduced LVOT gradients and symptomatic relief [29]. These agents are discussed in greater detail later in this review.

Device therapy is also essential for selected patients. Implantable cardioverter-defibrillator implantation is recommended for secondary prevention after cardiac arrest or sustained ventricular tachyarrhythmias, and for primary prevention in those with high-risk features, such as massive hypertrophy, family history of sudden cardiac death, or non-sustained ventricular tachycardia [12]. Heart transplantation should be considered in rare cases of end-stage disease with severe systolic dysfunction [30].

## 3. Pathology of HCM

Asymmetric left ventricular hypertrophy is a hallmark macroscopic finding in primary sarcomeric HCM, in contrast to proportional hypertrophy in the pressure-overloaded heart, such as hypertensive heart and aortic stenosis. The increase in heart weight, a general pathological indicator of cardiac hypertrophy, is relatively milder in HCM than in dilated cardiomyopathy cases. Typical HCM is characterized by asymmetric septal hypertrophy (ASH), with a septal-to-free wall thickness ratio of 1.3:1 or greater. Around 70% of HCM cases represent LVOTO, and a pressure gradient of 30 mmHg or more causes systolic anterior movement of the mitral valve [31]. In addition, the historical Maron’s classification of HCM subtypes has clearly shown diverse patterns of wall thickening based on echocardiographic short-axis cross-sectional images: Type I (basal septal), Type II (whole septal), Type III (septal and anterolateral wall), and Type IV (apical) hypertrophy patterns [32]. Nevertheless, the clinical manifestations of HCM are closely related to the presence of LVOTO and locality of mural hypertrophy. Current clinical practice guidelines classify HCM into five subtypes: HOCM, non-obstructive HCM, mid-ventricular obstruction, apical HCM, and dilated-phase HCM (D-HCM) (Figure 1) [12]. HOCM, the prototypical form of HCM, was introduced by Braunwald in the 1960s as idiopathic hypertrophic subaortic stenosis [33]. Although there is no clear macroscopic histopathological classificatory scheme for HCM, it is generally described according to these clinical subtypes. However, these subtypes are based on clinical modality observations synchronized with the cardiac cycle; thus, postmortem autopsy findings do not necessarily correspond to clinical recognition, and each ventricular cross-sectional slice may represent different wall thickening areas [34]. It is important to describe the characteristics of each individual case carefully.

The abnormal structure of HCM is not limited to ventricular wall thickening. Chronic contact injury of the endocardium below the aortic valve with systolic anterior movement and high blood flow velocity at the LVOTO cause endocardial thickening [31]. Abnormal morphology of the papillary muscle and chordae tendineae are also involved in LVOTO and mitral regurgitation [31,35]. Pathological ischemic scars are commonly associated with thickened ventricular walls (Figure 3A) [31]. Mural thrombi can be a problem in cases of D-HCM and when apical aneurysms are formed because of mid-ventricular obstruction (MVO) [36]. Whole-heart remodeling with various structural abnormalities should be considered as a treatment strategy for HCM.

The histological features have been described as bizarre myocardial hypertrophy with disorganization (BMHD) (Figure 3B) [12,37]. Bizarre three-dimensional disarrangement of muscular fascicles is often associated with bizarre nuclear morphology in markedly hypertrophic cardiomyocytes, the histopathological details of which were first introduced by Teare in 1958 based on eight autopsy cases in young adults, including seven cases of sudden death [1]. While the normal transverse diameter of myocardial cells is up to 15 µm in the right ventricle and approximately 18 µm in the left ventricle, that of cells in HCM often increases to over 30 µm [31]. Bizarre nuclear morphology similar to that of atypical tumor cells, such as nuclear enlargement, irregularity, variation in nuclear size, and dark staining of chromatin, is thought to reflect an increase in the amount of DNA in the damaged myocardium [38], which is a characteristic cellular finding not only in HCM but also in advanced cardiomyopathies. Hypertrophied myocardium is often accompanied by myofibrillar loss and, in severe cases, vacuolar degeneration. Disarray is an abnormal arrangement that shows multidirectional and complex branching with contact between the cardiac muscle fibers, often appearing as pinwheels, storiforms, or herringbone patterns [31]. Myocardial disarray is not specific to HCM, and may be physiologically seen in areas where muscle bundles of both ventricles intersect and around blood vessels; it is also observed in other diseases that cause cardiac hypertrophy, such as congenital heart diseases and hypertension. However, disarray is observed more frequently and widely and has a high diagnostic value in HCM [39]. Endomyocardial fibrous thickening is often associated with myocardial hypertrophy, and reactive fibroses of perivascular or interstitial patterns are also observed to various degrees. Additionally, plexiform fibrosis can be observed along the myocardium, showing a disorganized fascicular arrangement (Figure 3C) [12]. Replacement fibrosis suggesting an ischemic scar is also observed in relation to microvascular disease and myocardial oxygen supply–demand imbalance, which is detectable by cardiac MRI as well [40]. Small intramural coronary arteriole dysplasia (SICAD) is characterized by disorganized smooth muscle proliferation, resulting in microvascular disease (Figure 3C, inset) [12,40]. Approximately 15% of HCM cases progress to advanced myocardial fibrosis with retrograde morphology, indicating end-stage (burnout) cardiomyopathy (Figure 3D) [12,31].

Endomyocardial biopsy, invented in Japan in 1962, is the only method to obtain information about the cardiac tissue in vivo. It is essential for the follow-up examination of transplanted hearts and is useful for diagnosing myocarditis and cardiomyopathies [12,37,41]. As myocardial disarray, a histological characteristic of HCM, occurs most frequently in the middle layer of the ventricular wall [12,31,39], nonspecific endomyocardial biopsy findings obtained by sampling tissue from the subendocardium are common (Figure 3E). Rather, the value of EMB in suspected HCM cases is in its ability to differentiate among secondary cardiomyopathies such as amyloidosis and Anderson–Fabry disease [12,31,41,42]. In addition, disarray in HCM may be observed at the myofiber-level using electron microscopy (Figure 3F) [12,31]. It is difficult to predict genetic abnormalities such as *MYH7*, also known as β-myosin heavy chain, and *MYBPC3* from histological findings, and the observed histological findings are modified by various modifier genes, medical history of each patient, and environmental factors in addition to the influence of single sarcomere gene abnormality [2]. The histology of LVOTO in HCM is also seen in the surgical materials of septal myectomy, which often show severe myocardial degeneration with compensatory hypertrophy of cardiomyocytes, myofibrillar loss, fascicular disruption, endomyocardial thickening, and advanced fibrosis of interstitial and replacement patterns (Figure 3G,H) [43]. Myosin inhibitors have been shown to suppress the histogenesis of myocardial hypertrophy, disarray, and fibrosis in HCM mouse models [4]. In most HCM cases, histological examination in daily pathological practice may not be performed in its early stages. Therefore, it should be considered that HCM pathology is possibly associated with more complex and advanced retrograde degeneration than typical myocardial hypertrophy and disarray. Changes in human HCM pathology following administration of myosin inhibitors remain a topic for future investigation.

## 4. History of Myosin Inhibitor Development

Myosin inhibitors have been developed as innovative agents that can fundamentally modify HCM pathogenesis. The key milestones in their development are outlined below (Table 1).

### 4.1. Elucidation of the Molecular Mechanisms of HCM (1950s–2000s)

HCM was first described pathologically by Teare [1], who reported a myocardial disease characterized by asymmetric hypertrophy, particularly of the left ventricular myocardium, in autopsy cases of young individuals who died suddenly. One of the patients had a family member with a similar cardiac condition. In 1964, Braunwald et al. reported on several families affected by HCM and further clarified its familial nature [52]. In 1973, Clark et al. conducted echocardiographic studies on relatives of patients with HCM and demonstrated that the disease followed an autosomal dominant inheritance pattern with high but variable penetrance [44]. In 1990, mutations in the β-cardiac myosin heavy chain gene were identified as the first genetic cause of HCM, and co-segregation of these mutations with the disease phenotype confirmed the autosomal dominant mode of inheritance [45,46]. Since then, numerous causative genes have been identified and HCM is now primarily recognized as sarcomeric cardiomyopathy. Currently, first-line genetic testing panels include eight core sarcomere genes, *MYH7*, *MYBPC3*, *TNNI3*, *TNNT2*, *TPM1*, *MYL2*, *MYL3*, and *ACTC1*, with disease-causing mutations detected in approximately 30% of all-comers with HCM and >60% of patients with a family history of HCM [53]. By the 2000s, it had become increasingly evident that HCM was largely driven by hypercontractility and increased ATPase activity resulting from mutations in the sarcomere genes [3]. These insights highlighted the limitations of conventional treatments such as the use of beta blockers and calcium channel blockers in addressing the underlying disease mechanism.

### 4.2. Conceptual Shift Toward Myosin Modulation (2010s–Present)

In the early 2010s, targeting the cardiac myosin ATPase emerged as a promising therapeutic strategy for treating heart failure. In 2011, Cytokinetics reported omecamtiv mecarbil, a small molecule that enhances cardiac myosin ATPase activity, as a potential treatment for systolic heart failure [47]. Around the same time, Cooke et al. proposed that the biphasic kinetics of ATP turnover in cardiac myosin reflects two functional states: the disordered relaxed state (DRX) and super relaxed state (SRX) [54,55,56,57]. Excessive activation of myosin in the DRX state is believed to be the pathological hallmark of HCM [58,59]. In 2016, Green et al. reported that mavacamten (MYK-461)—a selective cardiac myosin ATPase inhibitor—attenuated ventricular hypertrophy, myocyte disarray, and fibrosis in mouse models carrying human HCM mutations [4]. It was also shown to downregulate hypertrophic and profibrotic gene expression [4].

### 4.3. Clinical Advancement of Myosin Inhibitors

In 2019, the PIONEER-HCM phase 2 trial showed that mavacamten significantly reduced LVOTO, improved exercise capacity (peak oxygen consumption [pVO_2_]), and alleviated symptoms in patients with HOCM [48]. In 2020, the EXPLORER-HCM trial—a phase 3, randomized, double-blind, placebo-controlled study—demonstrated that mavacamten significantly improved the pVO_2_ and NYHA functional class compared with the placebo. It also effectively reduced the LVOT gradient and improved symptom burden [49]. In April 2022, mavacamten was approved by the U.S. Food and Drug Administration (FDA) for the treatment of HOCM.

In 2021, Chuang et al. reported the discovery of aficamten (CK-274), a next-generation cardiac myosin inhibitor [5]. In a phase 1 clinical trial, aficamten demonstrated a human half-life consistent with pharmacokinetic predictions and achieved steady-state plasma concentrations within a two-week dosing window. In 2023, the REDWOOD-HCM phase 2 trial showed that aficamten significantly reduced LVOT gradients in patients with HOCM [60]. In 2024, the SEQUOIA-HCM phase 3 trial showed that aficamten improves the peak VO_2_ in patients with HOCM [50].

In 2024, the HORIZON-HCM study, a phase 3 open-label trial, reported that mavacamten was associated with improvements in LVOT gradients, cardiac biomarkers, and symptoms in Japanese patients with HOCM, which was comparable to the results of the EXPLORER-HCM trial [51]. In 2025, the Pharmaceuticals and Medical Devices Agency (PMDA) of Japan approved mavacamten for the treatment of HOCM.

### 4.4. Future Developments

Myosin inhibitors are now being investigated for use in non-obstructive HCM and in patients with coexisting heart failure and preserved ejection fraction. Long-term safety evaluations of the currently available agents are ongoing, and the development of shorter-acting myosin inhibitors is underway. Siontis et al. reported that artificial intelligence (AI)-based analysis of digital data from standard 12-lead electrocardiography is longitudinally correlated with changes in echocardiographic and laboratory markers during mavacamten treatment in patients with HOCM [61]. These findings suggest that AI-enhanced electrocardiography may be a useful tool for monitoring therapeutic responses in this population. Furthermore, as precision medicine advances through genetic profiling, individualized treatment approaches tailored to specific sarcomere mutations are expected to become increasingly feasible.

## 5. Myosin Regulation and Inhibition in HCM

To understand the mechanism of action of myosin inhibitors, it is essential to first understand the physiological interaction between myosin and actin, molecular mechanisms underlying sarcomere contraction, and pathological alterations of these processes in HCM.

### 5.1. Molecular Mechanism of Sarcomere Contraction

Cardiac contractions arise from the cyclical interaction between actin and myosin. Under low intracellular calcium concentrations ([Ca^2+^]), tropomyosin blocks the myosin-binding sites on actin filaments [62]. The troponin complex, which is composed of subunits C, I, and T, regulates this interaction. Troponin T anchors the complex to tropomyosin, whereas troponin I maintains tropomyosin in its blocking position on actin, thereby preventing access to the myosin binding sites on the actin surface. When [Ca^2+^] is elevated, calcium binds to troponin C, inducing a conformational change that weakens the troponin I–actin interaction and causes tropomyosin to shift laterally. This tropomyosin shift exposes the binding sites on actin, allowing the myosin heads to bind to actin (Figure 4A).

Myocardial contraction occurs through the binding of ATP to the myosin head, hydrolysis of ATP, and phosphate release. This process causes structural changes in the myosin head, generating the power stroke and resulting in the actin and myosin filaments sliding past one another (Figure 4B) [63]. The ATP-bound myosin head can only bind weakly to the actin filament (weakly bound state). As ATP is hydrolyzed to adenosine diphosphate (ADP) and inorganic phosphate (Pi), the myosin head can bind to the actin filament from the resting position to a strongly bound state. Subsequently, Pi is released from the myosin head, causing structural changes in the myosin head. This structural change causes the power stroke, and ADP is released from the myosin head. When a new ATP binds to the myosin head, it is released from the actin filament and returns to the weakly bound state (Figure 4C) [64,65]. This cyclical process underlies force generation and cardiac muscle contraction.

### 5.2. Myosin Head Conformations

Myosin molecules consist of two heavy chains, each with a motor domain (comprising the head and neck) and a tail domain. Myosin heads can adopt distinct conformational and biochemical states: the super-relaxed state (SRX), disordered relaxed state (DRX), and active state [54,55,56,57]. In the SRX, both the heads fold back against the thick filament backbone in a conformation known as the interacting-heads motif (IHM) [66,67], which is characterized by minimal ATPase activity. In the DRX, one head is released and assumes a primed conformation, partially prepared for actin engagement and ATP hydrolysis. In the active state, both heads disengage from the IHM and interact with the actin filaments to generate a contraction force (Figure 5) [58].

### 5.3. Conformations of Myosin Heads in HCM

HCM is characterized by left ventricular hypertrophy, hypercontractility, and impaired relaxation. HCM is primarily associated with pathogenic variants in more than 10 sarcomeric proteins, among which the β-myosin heavy chain (encoded by *MYH7*) and MYBPC3 (encoded by *MYBPC3*) are the major contributors [2,12]. These variants destabilize the SRX/IHM conformation, shifting the equilibrium toward the DRX and active states. As a result, the number of myosin heads available for actin binding increases, leading to increased ATP consumption, enhanced contractility, diastolic dysfunction, and pathological myocardial remodeling, including cardiomyocyte hypertrophy, disarray, and fibrosis [58,68].

Given the mechanism underlying sarcomere contraction described above and the pathophysiological role of SRX destabilization in HCM, strategies aimed at decreasing [Ca^2+^], decreasing [Ca^2+^] sensitivity, and stabilizing the SRX state are therapeutic targets for HCM. A small molecule called a direct myosin inhibitor was identified through screening to reduce ATPase activity, thereby decreasing the contractility force and promoting myocardial relaxation [4,69].

### 5.4. Mechanism of Action of Myosin Inhibitors

Myosin inhibitors are selective allosteric inhibitors of cardiac β-myosin ATPase. These compounds, exemplified by mavacamten and aficamten, bind to the myosin head and stabilize the SRX conformation by reinforcing the IHM structure (Figure 5) [70,71,72]. In doing so, they shift the balance from the DRX, which is available for actin interaction, to the energy-conserving SRX state, reducing the number of myosin heads able to participate in cross-bridge formation.

By promoting SRX states, myosin inhibitors decrease ATP turnover, actin–myosin cross-bridging, and contractile force generation, without altering intracellular [Ca^2+^] or β-adrenergic signaling pathways [69].

Biochemical and kinetic studies demonstrated that mavacamten decreases myosin ATPase activity and Pi release, and reduces the number of myosin-S1 heads that can interact with the actin thin filament during the transition from the weakly to the strongly bound state [70]. Mavacamten reinforces the autoinhibited state unique to two-headed cardiac myosin, enhancing autoinhibition of ATP turnover, ADP release, and lever-arm rotation [71].

Structural studies, including X-ray diffraction and cryo-electron microscopy also revealed that mavacamten promotes the SRX state [69].

In HCM mouse models, mavacamten reversed hypertrophy, fibrosis, and myocyte disarray [4]. It also rescued relaxation deficits and reduced hypercontractility in mouse and human cardiomyocytes carrying *MYBPC3* mutations [68]. Clinically, these effects translate to reduced left ventricular outflow tract obstruction and improved diastolic function.

Aficamten binds to an allosteric site on the myosin catalytic domain distinct from mavacamten, preventing the conformational changes required for strong actin binding [72]. By limiting the recruitment of functional myosin heads, aficamten reduces sarcomere shortening and cardiac contractility in rat cardiac myocytes and in mice carrying the hypertrophic R403Q myosin mutation [72].

Myosin inhibitors represent a novel class of therapeutics targeting the fundamental biophysical machinery involved in cardiac contraction. By stabilizing the SRX state of myosin, myosin inhibitors offer a disease-modifying approach to HCM by directly correcting the underlying molecular dysfunction responsible for hypercontractility and pathological remodeling. As our understanding of sarcomere energetics deepens, further refinements in myosin modulation may offer new avenues for treating a range of cardiomyopathies beyond HCM [73,74].

## 6. Cardiac Myosin Inhibitors and Therapeutic Evidence for HOCM

As mentioned above, beta blockers are used as the first-line therapy for LVOTO, and non-dihydropyridine calcium channel blockers are selected in cases where beta blockers are ineffective or not tolerated. Class 1a antiarrhythmic drugs (disopyramide or cibenzoline) are also expected to be effective; however, septal reduction therapies (SRTs) such as myocardial resection or percutaneous septal myocardial ablation (PTSMA) are indicated when the symptoms associated with LVOTO are difficult to control pharmacologically. In terms of the current state of treatment in Japan, baseline enrolment data from a prospective Japanese registry study of patients with HCM currently being conducted show that among 530 patients with HOCM, 386 (73%) had received oral medications only [14]. Among those treated with SRTs, 114 patients (22%) underwent PTSMA and 33 (6%) underwent myectomy. Regarding the status of LVOTO at registration after oral medication and/or SRT among the overall HOCM population, 63% of the patients treated with medication alone still exhibited a significant pressure gradient. In contrast, 21% of the patients who underwent PTSMA had LVOTO after PTSMA, and only 12% of the patients who underwent myectomy showed residual obstruction. Thus, before cardiac myosin inhibitors became available for use, adequate control of LVOTO often required SRT. This section describes the types of cardiac myosin inhibitors and their therapeutic effects on HOCM, with evidence from phase 3 trials (Table 2).

### 6.1. Mavacamten

The first-in-class cardiac myosin inhibitor—mavacamten—a drug specifically effective against HCM, was introduced against this background. Mavacamten has a variable terminal half-life that mainly depends on cytochrome P450 (CYP)2C19 metabolic status and has a long human half-life of approximately 7–9 days, typically requiring approximately six weeks to reach steady-state concentrations [77]. Table 2 shows the results of phase 3 trials of mavacamten in patients with HOCM.

In the EXPLORER-HCM trial, 251 adult patients with HOCM (LVOT pressure gradient, ≥50 mmHg; New York Heart Association [NYHA], class II or III symptoms) were assigned to receive either mavacamten or placebo for 30 weeks [49]. The primary endpoint was a composite measure of clinical response at week 30 compared with baseline, defined as an increase in peak oxygen consumption (pVO_2_) of ≥1.5 mL/kg/min and at least one NYHA class improvement, or an increase in pVO_2_ of ≥3.0 mL/kg/min without NYHA class worsening. The secondary endpoints included changes from baseline to week 30 in the post-exercise LVOT gradient, pVO_2_, proportion of NYHA class improvement, and patient-reported outcomes, including the Kansas City Cardiomyopathy Questionnaire (KCCQ) score. The primary endpoint was achieved in 37% and 17% of the patients in the mavacamten and placebo groups, respectively (*p* = 0.0005). Further, the increases in pVO_2_ and improvements in at least one NYHA class were also significantly greater in the mavacamten group than in the placebo group. Regarding the reduction in the LVOT pressure gradient, which is thought to be the main cause of improvement in this pathophysiology, the post-exercise pressure gradient was 36 mmHg lower in the mavacamten group than in the placebo group (*p* < 0.0001). Patients who were administered mavacamten demonstrated consistent benefits with respect to the primary endpoint across the prespecified subgroups, regardless of age, sex, beta blocker use, and whether or not pathogenic variants in the causative gene were detected. The NT-proBNP and high-sensitivity cardiac troponin I (hs-cTnI) levels, which were measured as exploratory endpoints, also showed significant decreases in the mavacamten treatment group [49].

In a subanalysis performed on the degree of improvement in patients’ health status as assessed by the KCCQ overall summary (KCCQ-OS) score [78], the change in score at 30 weeks was greater in the mavacamten group than in the placebo group (mean score 14.9 [standard deviation (SD), 15.8] versus 5.4 [13.7]; least-squares mean difference, +9.1 [95% confidence interval (CI), 5.5–12.8]; *p* < 0.0001). A marked improvement (≥20 points) in the KCCQ-OS score was observed in 36% of the patients in the mavacamten group versus 15% in the placebo group, yielding a number-needed-to-treat (NNT) of five. Structural changes in the heart following mavacamten treatment have been previously reported. An echocardiography study showed that the interventricular septal wall thickness increased by 1.4 mm in the placebo group after 30 weeks of treatment, whereas no increase in septal wall thickness was observed in the mavacamten group [79]. The left atrial volume index (LAVI) was also significantly reduced in the mavacamten group compared with that in the placebo group. Similar structural changes were confirmed by cardiac MRI, with the left ventricular mass index (LVMI), maximum wall thickness, and LAVI being significantly lower in the mavacamten group than in the placebo group [80].

The subsequent phase 3 VALOR-HCM study evaluated the proportion of patients eligible for SRT after 16 weeks of drug treatment in 112 symptomatic patients with HOCM (LVOT pressure gradient, ≥50 mmHg) who met the criteria for SRT [76]. After 16 weeks, 17.9% and 76.8% of the patients in the mavacamten and control groups, respectively, were eligible for SRT, which was a significant difference. These results suggest that mavacamten can be used not only to improve symptoms in patients with HOCM, but also to avoid SRT. The efficacy of this drug in Asian populations has also been reported. The EXPLORER-CN study conducted in China was a randomized clinical trial that enrolled 81 patients with HOCM. The primary endpoint was improvement in the Valsalva LVOT pressure gradient from baseline to week 30 after administration; a similar significant difference favoring mavacamten was observed [75]. Subsequently, the results of the HORIZON-HCM study, a phase 3 trial conducted in Japan (an open-label, single-arm trial involving 38 patients), have been reported; the primary endpoint was an improvement in the post-exercise peak LVOT pressure gradient at week 30 (a mean decrease from baseline of 60.7 mmHg) [51]. No serious complications associated with mavacamten occurred in any of these phase 3 trials, likely due to the regular monitoring of LVEF by echocardiography and dose adjustments based on echocardiographic findings.

### 6.2. Aficamten

Aficamten is a second-generation myosin inhibitor. It has a median half-life of 99.6 h, and a stable blood concentration is reached approximately two weeks after the start of administration [5], which has the advantage of allowing the drug dosage to be increased or decreased quickly. In the phase 3 SEQUOIA-HCM trial (Table 2), 282 adult patients with HOCM (NYHA, class II or III; pVO_2_ < 90%; LVOT pressure gradient ≥ 30 mmHg at rest and ≥50 mmHg after the Valsalva maneuver) were assigned to receive either aficamten or placebo for 24 weeks [50]. The primary endpoint was the change in pVO_2_ from baseline to week 24; ten pre-specified secondary endpoints were defined, including the degree of reduction in the LVOT pressure gradient. The mean change in pVO_2_ was 1.8 mL/kg/min (95% CI, 1.2–2.3) and 0.0 mL/kg/min (95% CI, −0.5 to 0.5) in the aficamten and placebo groups (least-squares mean between-group difference, 1.7 mL/kg/min; 95% CI, 1.0–2.4; *p* < 0.001), respectively.

Sub-analysis of this primary endpoint revealed consistent benefits of aficamten, regardless of age, sex, beta blocker use, and the presence of pathogenic genetic variants. Significant improvements were observed in all ten secondary endpoints—including the LVOT pressure gradient and NYHA class—in the aficamten group compared with the placebo group. Biomarker analysis measured the NT-proBNP and hs-cTnI levels [81]. At week 8, aficamten treatment produced a reduction in both biomarkers that was nearly the same as that observed later in the study: NT-proBNP was reduced by 79% (95% CI 76–83%, *p* < 0.001) and hs-cTnI by 41% (95% CI 32–49%, *p* < 0.001). The reduction in biomarkers persisted until week 24, when the relative reduction from baseline was 80% (95% CI 77–83%, *p* < 0.001) for NT-proBNP and 43% (95% CI 36–49%, *p* < 0.001) for hs-cTnI. The NT-proBNP and hs-cTnI levels returned to their baseline values after the completion of the treatment period and cessation of aficamten treatment. Reductions in the levels of these two biomarkers at 24 weeks were strongly associated with improvements in the LVOT pressure gradient, KCCQ score, and pVO_2_. Similar to the results for mavacamten mentioned above, evaluation using the KCCQ showed that 29.7% of patients in the aficamten group achieved a significant improvement in the KCCQ-OS score (≥20 points), compared with 12.4% in the placebo group, yielding an NNT of 5.8 [82]. Regarding the impact of aficamten on the echocardiographic cardiac structure of patients with HOCM, aficamten significantly lowered the resting and Valsalva LVOT gradients over 24 weeks and decreased the ventricular wall thickness, LVMI, and LAVI [83]. These structural changes have also been confirmed by cardiac MRI [84].

### 6.3. Mechanisms Underlying the Therapeutic Effect of Cardiac Myosin Inhibitors on HOCM

Mavacamten and aficamten have been shown to improve exercise tolerance, symptoms, biomarker levels, and cardiac remodeling in patients with HOCM. The following pathological changes are believed to underlie these therapeutic effects: (1) LVOTO relief and (2) direct improvements in myocardial energy efficiency and diastolic function. LVOTO relief is thought to play a major role in improving parameters such as pVO_2_, NYHA class, KCCQ score, and NT-proBNP and cardiac troponin I levels, in patients with HOCM. However, it is important to consider the effects of CMI treatment on cardiac structural changes (remodeling). Notably, both drugs were reported (somewhat unexpectedly) to suppress the increases in LV wall thickness and reduce LVMI and LAVI [80,84]; however, it remains unclear whether these effects are direct consequences of myosin–actin binding inhibition and reduced cross-bridge formation due to cardiac myosin inhibitors. Several studies have reported structural changes in the heart after SRT [85,86,87]. It is known that the LVOTO relief by PTSMA suppresses myocardial remodeling, including regression of left ventricular hypertrophy and reductions in LVMI and LAVI, as demonstrated by echocardiography and cardiac MRI [86,87]. Therefore, it cannot be ruled out that the structural changes were primarily attributable to LVOTO relief, that is, to a reduction in the afterload. Further insights into these therapeutic effects are expected from the results of an ongoing phase 3 trial in patients with non-obstructive HCM.

## 7. Myosin Inhibitors in HCM: Current Landscape and Future Perspectives

HCM is a complex genetic cardiac disorder characterized by left ventricular hypertrophy. HOCM, characterized by the presence of an LVOT gradient, poses therapeutic challenges, particularly in patients who remain symptomatic despite optimal medical treatment. SRT, including surgical myectomy (SM) and alcohol septal ablation (ASA), are cornerstone therapies for gradient relief [88]. However, several lines of evidence indicate the efficacy and safety of CMIs, a novel class of agents targeting the underlying hypercontractile phenotype of the disease [89]. Despite the promising results of phase 3 trials, several limitations and unsolved issues persist regarding their long-term efficacy, safety, and applicability across HCM subtypes (Figure 6). This review summarizes the current evidence on CMIs in HCM, focusing on the boundaries of the present knowledge and outlining key areas for future investigation.

### 7.1. Lack of Long-Term Outcomes of Mavacamten Compared with Septal Reduction Therapies in HOCM

Early phase and pivotal trials have confirmed the short-term efficacy of mavacamten in HOCM. In the EXPLORER-HCM study, 37% of patients receiving mavacamten achieved the primary composite endpoint of improvements in the pVO_2_ and NYHA functional class over a 30-week period, compared with 17% in the placebo group [49]. Similarly, the VALOR-HCM trial demonstrated a marked reduction in the proportion of patients requiring SRT after 16 weeks of therapy (18% vs. 77% with placebo) [76]. However, both the trials were limited by their short follow-up periods. To address this issue, the MAVA-LTE study—the open-label extension of the phase 3 EXPLORER-HCM study—followed 231 patients for a median of 166.1 weeks [90]. The treatment retention rate was high (91.3%), with a cumulative exposure of 739 patient-years, the longest reported to date in a mavacamten clinical trial. At week 180, 77.9% of the patients had improved by at least one NYHA class, and the EQ-5D-5L index score increased by 0.09 points on average, indicating an improved general health status. Resting and Valsalva LVOT gradients were reduced by 40.3 mmHg and 55.3 mmHg, respectively; 82.7% of patients exhibited Valsalva gradients below 30 mmHg. The NT-proBNP levels decreased from a median of 766 to 118 ng/L. Importantly, 20 patients (8.7%) experienced transient reductions in the left ventricular ejection fraction (LVEF) to <50%, all of which resolved with temporary treatment interruption. Two patients (0.9%) underwent SRT after treatment discontinuation; however, neither case was attributable to the lack of efficacy of mavacamten. None of the patients had an LVEF < 30%. Five deaths (2.2%) occurred during the study period; all of them were deemed unrelated to mavacamten treatment. While the study confirmed acceptable long-term safety, with approximately 96% of the 1870 reported adverse events classified as mild or moderate in severity, its limitations include a relatively small sample size, an open-label design, and the absence of a control group.

In contrast, robust long-term outcomes for SRT have been reported by Maurizi et al., who studied 1832 patients—including 124 pediatric cases—undergoing SM (*n* = 1377) or ASA (*n* = 455), with a median follow-up of 6.8 years (interquartile range, 3.4–9.8 years [range, 1–54]) [91] Post-procedural resting gradients declined from 77 to approximately 10 mmHg. Freedom from events with heart failure and ventricular arrhythmia at 10 years was 83%, and the overall long-term all-cause mortality rate was 12% (9% for SM and 18% for ASA). In addition, the procedural mortality at 30 days was low (0.4% overall; 0.36% for SM; 0.66% for ASA), highlighting the procedural safety in experienced centers. The longest follow-up period was 54 years in adults, offering unique insights into the durability of outcomes across patient lifespan.

Collectively, these findings illustrate the expansion of the therapeutic options for HOCM. Mavacamten offers a non-invasive, titratable option with growing evidence of short- and intermediate-term efficacy. Nevertheless, extensive and well-validated data supporting SRT, particularly regarding survival and durability, remains the standard by which newer therapies can be compared. A key limitation of mavacamten is the lack of long-term outcomes and real-world data comparable to those available for SRT, particularly in terms of procedural durability and survival over decades. The current data regarding younger and middle-aged patients, for whom long-term outcomes and disease stability are paramount, do not yet justify broadly replacing SRT with mavacamten. However, ongoing real-world investigations, most notably the DISCOVER-HCM registry (NCT05489705), are expected to shed light on the long-term effectiveness, safety, and treatment patterns of mavacamten in routine clinical practice, thereby potentially narrowing this critical evidence gap [92]. Continued prospective, randomized, and long-term comparative studies are essential to guide individualized treatment planning in this evolving clinical landscape.

### 7.2. Myosin Inhibitors vs. Beta Blockers as a First-Line Therapy in HOCM

Beta blockers are currently used as first-line therapy in patients with symptomatic HOCM. Consequently, pivotal phase 3 studies were conducted to evaluate the effects of CMIs as add-on agents in patients already receiving beta blockers. Therefore, the role of CMIs as a first-line therapy for HOCM remains unclear. To resolve this issue, the ongoing phase 3 MAPLE-HCM trial is the first randomized head-to-head comparison of a CMI (aficamten) with a beta blocker (metoprolol) as initial monotherapy in treatment-naïve patients with symptomatic HOCM [93]. This study aimed to demonstrate the relative effectiveness and tolerability of the two drug classes by evaluating outcomes such as pVO_2_, NYHA functional class, KCCQ-CSS, biomarker levels, and cardiac remodeling measures over 24 weeks.

CMIs exert their effects by mitigating excessive contractility through the selective reduction of actin–myosin cross-bridge formation. This targeted action allows for the normalization of systolic function without significant effects on heart rate or blood pressure. In contrast, beta blockers act indirectly through negative inotropic and chronotropic effects. These distinctions raise the possibility that CMIs may offer superior disease-modifying benefits, and the MAPLE-HCM trial may help determine their suitability as a first-line therapy.

### 7.3. Myosin Inhibitors in Non-Obstructive HCM: Promises and Pitfalls

The MAVERICK-HCM trial, a phase II double-blind, placebo-controlled study enrolling 59 patients with symptomatic non-obstructive HCM (NYHA class II–III, LVEF ≥ 55%), demonstrated that mavacamten significantly reduced the circulating levels of NT-proBNP and cardiac troponin I, suggesting a reduction in myocardial wall stress and injury [94]. However, no significant improvement was observed in the pVO_2_ or NYHA functional class in the overall intention-to-treat population. Notably, five participants experienced asymptomatic reductions in LVEF to ≤45%, all of which were reversible after drug discontinuation, indicating the potential benefit and safety in non-obstructive HCM. Based on these findings, the phase 3 ODYSSEY-HCM trial was designed to evaluate the efficacy and safety of mavacamten in a larger population of patients with symptomatic non-obstructive HCM [95]. This double-blind, randomized, placebo-controlled trial enrolled participants with NYHA class II–III symptoms and LVEF ≥ 60%; the participants were randomized 1:1 to receive either mavacamten or placebo. The primary endpoints were changes from baseline to week 48 in the KCCQ-CSS and pVO_2_ on cardiopulmonary exercise testing. However, according to a recent press release by Bristol Myers Squibb, the ODYSSEY-HCM trial did not meet its primary endpoint, as mavacamten failed to demonstrate a statistically significant improvement in the KCCQ-CSS and pVO_2_ compared with placebo.

In contrast, aficamten is a CMI with higher selectivity, shorter half-life, and more predictable pharmacokinetics than mavacamten. Preclinical studies have shown that aficamten allows for more rapid and reversible modulation of sarcomere contractility, which may be particularly advantageous in patients with preserved systolic function, such as those with non-obstructive HCM [5]. Preliminary evaluation of aficamten, a next-generation CMI, in patients with non-obstructive HCM was conducted in Cohort 4 of the REDWOOD-HCM trial, an open-label, exploratory sub-study involving 41 symptomatic patients (NYHA class II–III) with a resting LVOT gradient < 30 mmHg and LVEF ≥ 60% [96]. Aficamten was administered over a 10-week treatment period using individualized dose titrations based on echocardiographic parameters. Treatment with aficamten was associated with a reduction in NT-proBNP and troponin I levels, improvement in echocardiographic markers of diastolic function, and better patient-reported quality of life measures. Aficamten was generally well-tolerated, with no reports of serious adverse events related to its treatment. Importantly, three patients experienced a transient reduction in LVEF below 50%, all of which resolved promptly following drug discontinuation. However, this study lacked a control group, limiting any definitive conclusions regarding efficacy and safety. The ACACIA-HCM trial (NCT06081894) is currently underway to evaluate the therapeutic potential of aficamten. This global, randomized, double-blind, placebo-controlled phase III study will enroll adults with symptomatic non-obstructive HCM (NYHA class II–III, LVEF ≥ 60%). Participants will receive aficamten or placebo for 36 weeks, with the primary endpoints being the changes in KCCQ-CS and pVO_2_ from baseline. The results of the ACACIA-HCM trial are anticipated to provide definitive evidence regarding the clinical utility of aficamten in patients with non-obstructive HCM.

In summary, the effects of cardiac myosin inhibition in non-obstructive HCM remain unclear. Early phase studies have indicated favorable biological effects, and future large-scale trials are expected to determine whether these agents can translate into meaningful symptomatic and functional improvements in this challenging and underserved patient population.

### 7.4. Pharmacogenomics and Drug Metabolism: CYP2C19 and Clinical Implications

Mavacamten undergoes hepatic metabolism predominantly via CYP2C19, a cytochrome P450 enzyme system, followed by CYP3A4 and CYP2C9 [97]. Consequently, genetic polymorphisms that modulate CYP2C19 enzymatic activity can alter drug exposure and elimination profiles.

Individuals with poor metabolizer status—typically carrying two non-functional alleles such as *2/*2, *2/*3, or *3/*3—may experience markedly prolonged half-lives and elevated systemic exposure to drugs compared with normal metabolizers [77]. This elevation in systemic drug levels may be associated with an increased likelihood of excessive systolic function suppression, potentially predisposing these patients to heart failure.

The prevalence of poor metabolizer phenotypes varies substantially among ethnic groups. In East Asian populations, the estimated prevalence of poor metabolizer status is significantly higher than that observed in European or African cohorts [98].

Clinically, the co-administration of potent CYP2C19 or CYP3A4 inhibitors (e.g., fluvoxamine and fluconazole) is contraindicated owing to the risk of enhanced drug accumulation. Conversely, strong enzyme inducers (e.g., rifampicin, carbamazepine, phenytoin and St John’s wort) can reduce mavacamten exposure and compromise its efficacy. Even mild-to-moderate CYP modulation, such as that caused by omeprazole (CYP2C19) or diltiazem/verapamil (CYP3A4), warrants dose adjustments and closer monitoring. Food has no clinically meaningful effect on exposure, and mavacamten may be administered with or without meals. Owing to the narrow therapeutic window, periodic echocardiographic assessment during therapy is mandatory to ensure that adequate LVEF is maintained. Treatment should be temporarily suspended if the LVEF declines below the threshold value or if heart failure symptoms emerge. To support safe prescribing practices, mavacamten is distributed under a restricted-access program in the United States, requiring physician certification and adherence to defined monitoring protocols. In Japan, prescribers must complete an authorized e-learning course. For a comprehensive list of interacting agents and management recommendations, clinicians should consult the most recent full prescribing information before initiation and whenever starting or stopping concomitant drugs.

However, aficamten, a structurally distinct myosin inhibitor, may mitigate several of these concerns. Its metabolic pathway appears to be less reliant on polymorphic CYP enzymes and displays a shorter half-life, more predictable systemic exposure, and quicker attainment of a steady state [5]. These features may reduce the burden of pharmacogenomic screening and enable broader clinical applications with fewer pharmacogenomic constraints.

### 7.5. Genetic Heterogeneity and Treatment Responsiveness

Recent evidence suggests that mavacamten maintains its therapeutic benefits across various sarcomeric gene-variant subtypes in patients with symptomatic HOCM. In the VALOR-HCM trial, individuals carrying pathogenic or likely pathogenic variants—including *MYBPC3* (*n* = 13), *MYH7* (*n* = 3), *TNNT2* (*n* = 1), and *TNNI3* (*n* = 1)—as well as those with variants of uncertain significance or no identified sarcomeric mutations, exhibited similarly sustained improvements over 128 weeks. These benefits included reduction in LVOT gradients, symptomatic relief, and favorable shifts in biomarker profiles [99]. These data indicated that a known sarcomeric mutation may not be essential for achieving a clinical response.

However, HCM encompasses a genetically diverse population, and certain patients possess mutations in non-sarcomeric genes such as *PLN* or *ALPK3* [100]. Given that mavacamten exerts its effects by attenuating hypercontractility at the sarcomere level, its efficacy against non-sarcomeric subtypes remains questionable. Moreover, the number of individuals with such genotypes in clinical trials was small, indicating uncertainty regarding the applicability of mavacamten in these contexts. Additional investigations are warranted to clarify its therapeutic potential and safety in patients with non-sarcomeric or atypical genetic variants.

### 7.6. Future Directions

The development and clinical application of CMIs have enabled the development of mechanism-specific therapies for HCM. Although early clinical results, particularly for HOCM, have been encouraging, many challenges remain. These include limited long-term outcome data, inter-individual differences due to pharmacogenomic factors, potential off-target effects, and uncertain benefits in patients with non-sarcomeric or non-obstructive disease. Further clinical trials and real-world studies are warranted to address these issues.

## 8. Conclusions

Several guidelines for the management of cardiomyopathy [20] and heart failure [30] have highlighted the importance of broad phenotype-based approaches in providing disease-modifying therapies, including those for cardiac amyloidosis. Among these, CMIs directly address sarcomere hypercontractility and dysfunction in the pathogenesis of HCM. Based on the deliverables from experimental studies, clinical trials have demonstrated efficacy in reducing LVOTO and improving exercise capacity, along with a favorable safety profile, leading to strong guideline recommendations for its use in clinical practice. Several limitations and unresolved questions remain regarding its long-term durability, safety, and applicability across HCM subtypes as well as in non-HCM subtypes, such as heart failure with a preserved ejection fraction. However, this unique “bench-to-bedside” development path for CMIs provides additional critical information for determining the roles of these agents.

## Figures and Tables

**Figure 1 ijms-26-09347-f001:**
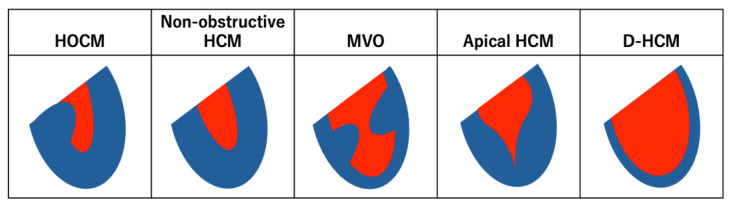
Subtypes of HCM. Subtypes of hypertrophic cardiomyopathy (HCM) are classified according to the pattern of left ventricular wall thickening observed in the long-axis cross-section, which closely reflects their underlying pathophysiological characteristics. In each diagram, the left ventricular wall is depicted in blue and the ventricular cavity in red. Abbreviations: HCM, hypertrophic cardiomyopathy; HOCM, hypertrophic obstructive cardiomyopathy; MVO, mid-ventricular obstructive HCM; D-HCM, dilated phase or end-stage HCM.

**Figure 2 ijms-26-09347-f002:**
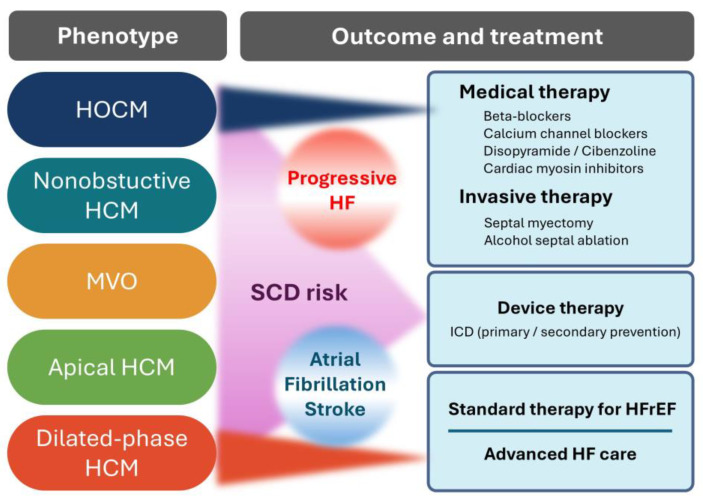
Clinical phenotypes, outcomes, and treatment strategies in hypertrophic cardiomyopathy (HCM). This schematic illustrates the major phenotypes of HCM, associated adverse outcomes (progressive heart failure, sudden cardiac death, atrial fibrillation/stroke), and representative treatment options, including medical therapy, septal reduction, device therapy, and advanced heart failure care. HCM, hypertrophic cardiomyopathy; HOCM, hypertrophic obstructive cardiomyopathy; MVO, mid-ventricular obstruction; SCD, sudden cardiac death; HFrEF, heart failure with reduced ejection fraction; HF, heart failure; ICD, implantable cardioverter-defibrillator.

**Figure 3 ijms-26-09347-f003:**
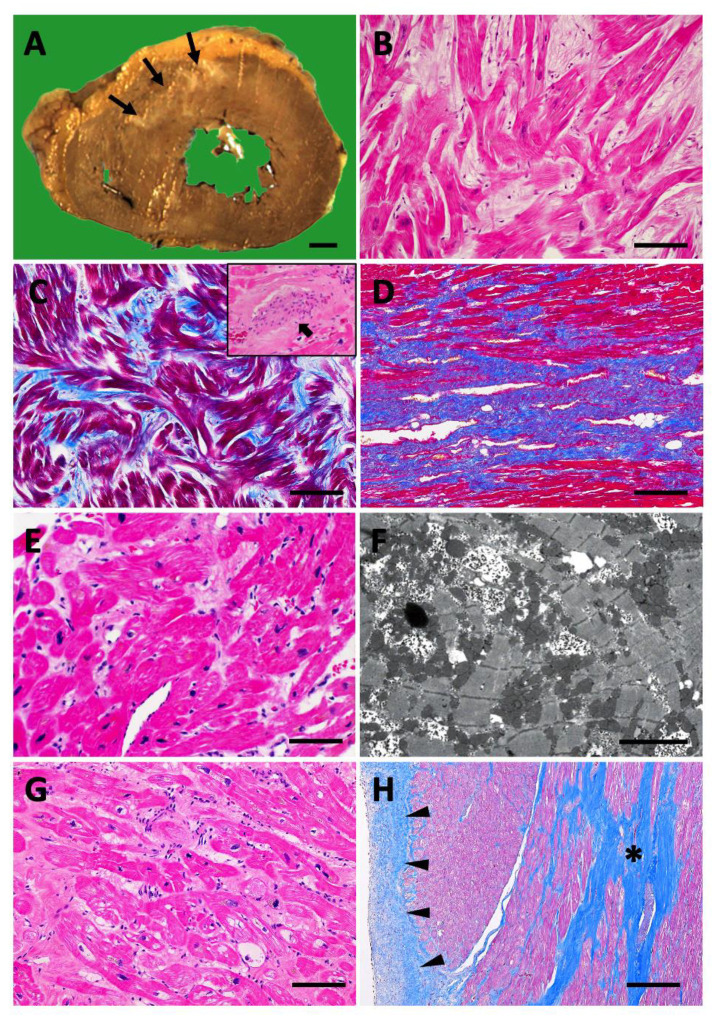
Histopathological characteristics of hypertrophic cardiomyopathy (HCM). (**A**–**D**) Autopsy findings. (**A**) Gross findings in the cross-section. Bar = 1 cm; arrows indicate the ischemic scar. (**B**) Histological findings of bizarre myocardial hypertrophy with disorganization (BMHD). Hematoxylin & eosin (HE) staining; Bar = 100 µm. (**C**) Plexiform fibrosis. Masson trichrome (MT) staining, Bar = 100 µm. Inset; arrow indicates small intramural coronary arteriole dysplasia (SICAD). (**D**) Advanced fibrosis in dilated phase of HCM (D-HCM). MT staining. Bar = 200 µm. (**E**,**F**) endomyocardial biopsy (EMB) findings. (**E**) Relatively characteristic histology with hypertrophic cardiomyocytes and fascicular disarrangement. HE staining, Bar = 100 µm. (**F**) Myofiber disarray. Transmission electron microscopy. Bar = 5 µm. (**G**,**H**) Histological findings in the surgical septal myectomy material. (**G**) Severe retrograde degeneration of cardiomyocytes at the left ventricular outflow tract obstruction (LVOTO). HE staining, Bar = 100 µm. (**H**) Endomyocardial thickening (area indicated by filled triangles) and replacement fibrosis (asterisk). MT staining. Bar = 200 µm. (**A**–**H**); Original artwork for this article.

**Figure 4 ijms-26-09347-f004:**
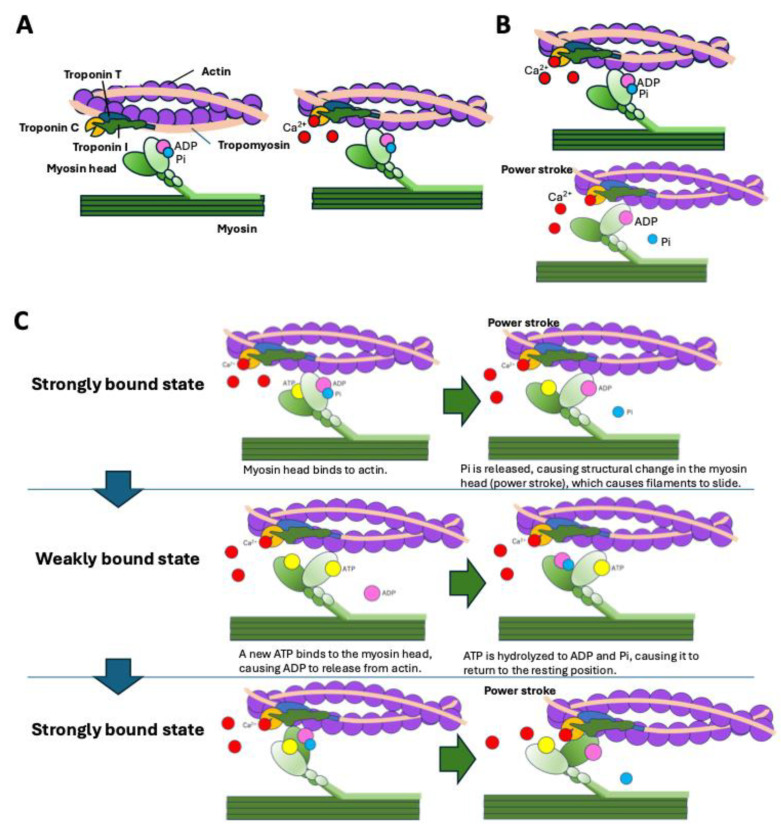
The mechanism of sarcomere contraction. (**A**) At low [Ca^2+^], tropomyosin blocks actin–myosin interaction. When the intracellular [Ca^2+^] increases and Ca^2+^ binds to troponin C, the conformational change in troponin C induces a tropomyosin shift, enabling actin–myosin binding. (**B**) Adenosine triphosphate (ATP) bound to the myosin head is hydrolyzed by the intrinsic ATPase activity of myosin into adenosine diphosphate (ADP) and inorganic phosphate (Pi). When Pi is released from the myosin head, it causes a structural change in the myosin head, generating the power stroke and resulting in the actin and myosin filaments sliding past one another. (**C**) The ATP-bound myosin head can only bind weakly to the actin filament (weakly bound state). As ATP is hydrolyzed to ADP and Pi, the myosin head can bind to the actin filament (strongly bound state). Subsequently, Pi is released from the myosin head, causing the power stroke. After the power stroke, ADP is released from the myosin head. When a new ATP binds to the myosin head, myosin is released from the actin filament and returns to the weakly bound state.

**Figure 5 ijms-26-09347-f005:**
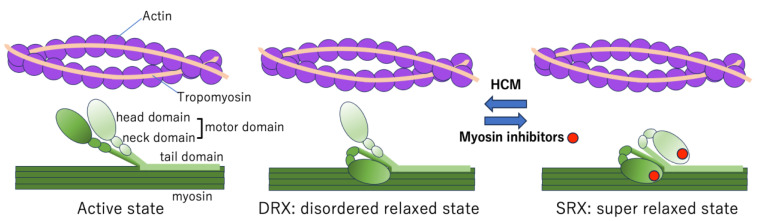
Myosin head conformations. Myosin molecules are composed of two identical heavy chains, each of which possesses a motor domain (head and neck domain) and a tail domain. Myosin heads can form distinct biochemical and structural states: the super-relaxed state (SRX), disordered relaxed state (DRX), and active state [58,59]. In the SRX, both heads fold back onto the tail, which significantly reduces ATP turnover. In contrast, the DRX allows one head to adopt a primed conformation. Transition to the active state occurs when both heads engage with actin filaments during contraction. In HCM mutations, the SRX structure is destabilized, shifting the equilibrium toward DRX and active states, thereby increasing the number of myosin heads available for binding to actin. Myosin inhibitors bind to the head domain and increase the SRX population, which reduces pathological contractility by decreasing actin–myosin cross-bridging.

**Figure 6 ijms-26-09347-f006:**
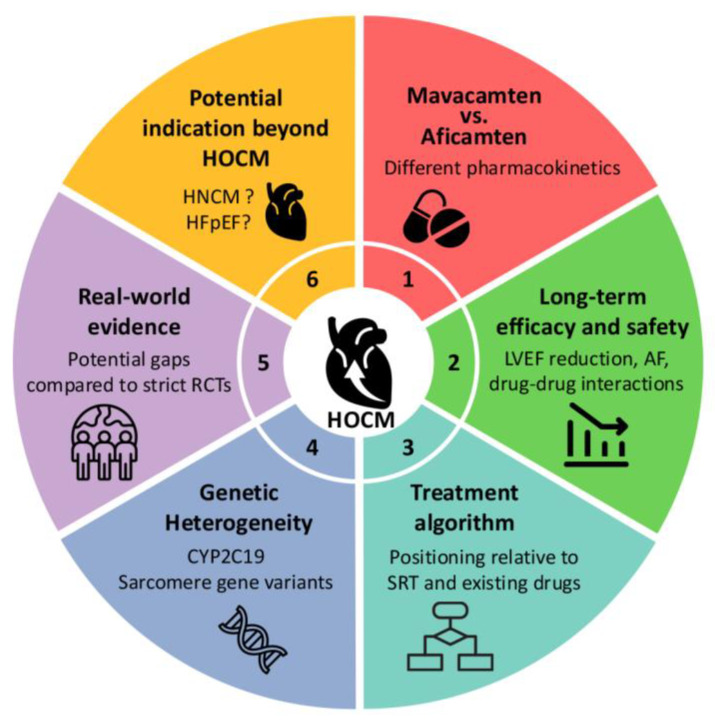
Key unanswered questions regarding the use of cardiac myosin inhibitors (CMIs) in hypertrophic cardiomyopathy (HCM). This schematic outlines six unresolved questions that remain despite favorable results of phase 3 trials testing the efficacy of mavacamten and aficamten in hypertrophic obstructive cardiomyopathy (HOCM). 1, it remains unclear whether these agents differ meaningfully in hemodynamic efficacy, symptom relief, or tolerability. 2, their long-term durability and safety—particularly the risk of sustained left-ventricular functional decline, arrhythmias, or progression of cardiomyopathy—have yet to be established. 3, the optimal sequencing of CMIs with established treatments such as beta blockers and septal reduction therapy (SRT) is still undetermined. 4, the extent to which genetic factors (e.g., CYP2C19 activity and sarcomeric variants) modulate their efficacy and toxicity is uncertain. 5, real-world data are needed to confirm the trial findings in broader clinical populations. 6, the efficacy and safety of CMIs in non-obstructive HCM remain unproven. Clarifying these gaps is essential to refine the therapeutic role of CMIs across the full spectrum of HCM.

**Table 1 ijms-26-09347-t001:** Key historical milestones in the development of myosin inhibitors.

Period	Milestone	Reference
1958	Teare reported the first pathological description of asymmetric septal hypertrophy in young sudden cardiac death cases.	[1]
1973	Familial clustering and autosomal dominant inheritance patterns of HCM were identified.	[44]
1990	*MYH7* mutations were identified as a genetic cause of HCM.	[45,46]
2000s	Hypercontractility and increased ATPase activity were established as key pathogenic mechanisms.	[3]
2011	Cytokinetics introduced omecamtiv mecarbil, an ATPase activator for systolic heart failure.	[47]
2016	Mavacamten (MYK-461), a cardiac myosin inhibitor was shown to reverse hypertrophy in mouse models of HCM.	[4]
2019	PIONEER-HCM trial: Mavacamten improved LVOTO and exercise capacity.	[48]
2020	EXPLORER-HCM trial: Demonstrated efficacy and safety of mavacamten in patients with HOCM.	[49]
2021	The cardiac myosin inhibitor, aficamten (CK-274), was discovered.	[5]
2022	U.S. FDA approved mavacamten for HOCM.	
2024	SEQUOIA-HCM phase 3 trial showed that aficamten improved peak VO_2_ in patients with HOCM.	[50]
2024	HORIZON-HCM trial in Japanese patients showed comparable efficacy.	[51]

ATPase, adenosine triphosphatase; FDA, food and drug administration; HCM, hypertrophic cardiomyopathy; HOCM, hypertrophic obstructive cardiomyopathy; LVOTO, left ventricular outflow tract obstruction; *MYH7*, myosin heavy chain 7.

**Table 2 ijms-26-09347-t002:** Phase 3 trials of cardiac myosin inhibitors in patients with HOCM.

Trial Name	EXPLORER-HCM [49]	EXPLORER-CN [75]	HORIZON-HCM [51]	VALOR-HCM [76]	SEQUOIA-HCM [50]
Drug	Mavacamten	Mavacamten	Mavacamten	Mavacamten	Aficamten
Trial location	Europe and USA	China	Japan	USA	Europe and USA
Eligible patients	NYHA II or III	NYHA II or III	NYHA II or III	Eligible for SRT (NYHA III or IV)	NYHA II or III
Number of patients	251	81	38	110	282
Study design	Double-blind randomized	Double-blind randomized	Open-label single-arm	Double-blind randomized	Double-blind randomized
Study period	30 weeks	30 weeks	30 weeks	16 weeks	24 weeks
Primary endpoint	pVO_2_ and NYHA class	Valsalva LVOT-PG	Post-exercise LVOT-PG	proceeding with SRT or eligible for SRT	pVO_2_
LVOT-PG	↓	↓	↓	↓	↓
pVO_2_	↑	Data not collected	Data not collected	Data not collected	↑
NYHA class	Improved	Improved	Improved	Improved	Improved
KCCQ	↑ (Improved)	↑ (Improved)	↑ (Improved)	↑ (Improved)	↑ (Improved)
NT-proBNP	↓	↓	↓	↓	↓
Cardiac troponin	↓	↓	↓	↓	↓
LVMI	↓	↓	↓	↓	↓
LAVI	↓	↓	↓	↓	↓

KCCQ, Kansas City Cardiomyopathy Questionnaire; LAVI, left atrial volume index; LVMI, left ventricular mass index; LVOT, left ventricular outflow tract; NT-proBNP, N-terminal pro-brain natriuretic peptide; NYHA, New York Heart Association; PG, pressure gradient; pVO_2_, peak oxygen consumption; SRT, septal reduction therapies.

## Data Availability

No new data were created or analyzed in this study. Data sharing is not applicable to this article.

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
