# Peer review of "Cardiac Myosin Inhibitors in Hypertrophic Cardiomyopathy: From Sarcomere to Clinic"

_ijms, 2025, doi:10.3390/ijms26199347_

Round 1

Reviewer 1 Report

Comments and Suggestions for Authors

The article by Nakamura and colleagues is a comprehensive review intended to introduce a new class of drugs, cardiac myosin inhibitors, for curing a range of heart diseases. The well-written review, based on voluminous literature data, provides extensive information about two of these medicines, mavacamten and aficamten, which have been tested in several clinical trials and shown effectiveness in treating hypertrophic cardiomyopathies (HCM).

The HCM types, methods of diagnostics and treatment, both traditional and modern, and comparison of the effects of several classes of therapeutic drugs to those of myosin inhibitors are considered. The review, with no doubt, will be of interest to a wide circle of physiologists, pharmacologists, and practicing physicians.

In my opinion, however, several issues need correcting.

  1. Section 5. (Mechanism of Action of Myosin Inhibitors) and paragraph 5.4. (the same title ?).

Despite the plural in the title, the information in this section relates to only one inhibitor, mavacamten. From the title, one can take the section is pretending to disclose the molecular action mechanism, whereas, in fact, it is currently being investigated. In fact, the information in the section rather concerns the effects of mavacamten, i.e., what it does, not how.

  1. Figure 3C. There is an obvious inconsistency between the termweakly bound stateand the image, where myosin heads are detached from actin. The term 'weakly bound' refers to a biochemical state; physically, these heads are attached to actin but incapable of making a power stroke (see e.g. Ferenczi et al., Structure, 13.1, doi: 10.1016/j.str.2004.11.007). So the figure has to be corrected.
  2. The term ‘ATPase’ (lines 362 and 372), as it is used there, is misleading, making an impression that this is an additional separate participant of the process, whereas ATPase is myosin itself.
  3. Fig. 4, the right panel. The biochemical SRX state, as correctly mentioned in the text, is structurally kept in the IHM, i.e., the two heads interact with each other (e.g., in Schmid and Toepfer in the refs. list or Craig and Padrón, J Gen Physiol.,2022, doi:10.1085/jgp.202113012). In the drawing, however, they are separated.

Lines 208, 209: 100 μm, not 100 mm.

Line 253: EBM? I guess it should be EMB (endomyocardial biopsy).

Line 354: …troponin I inhibits the actin–myosin interaction. Troponin I does not directly inhibit the interaction. It keeps tropomyosin bound to actin in the blocking state, thereby closing the myosin binding sites on its surface.

It is advisable to decode abbreviations CYP2C19, CYP3A4, CYP2C9, and NT-proBNP.

Author Response

Response to reviewer 1

We greatly appreciate the reviewer’s insightful comments.

Comment 1: Section 5. (Mechanism of Action of Myosin Inhibitors) and paragraph 5.4. (the same title ?). Despite the plural in the title, the information in this section relates to only one inhibitor, mavacamten. From the title, one can take the section is pretending to disclose the molecular action mechanism, whereas, in fact, it is currently being investigated. In fact, the information in the section rather concerns the effects of mavacamten, i.e.what it does, not how.                      

Response: We appreciate this valuable feedback. In accordance with the reviewer’s suggestion, we have revised the title of Section 5 to Myosin Regulation and Inhibition in HCM, and have added a description of aficamten in paragraph 5.4 (lines 465–469). Furthermore, we have expanded the section to include the mechanisms of both mavacamten and aficamten (lines 442-476).

Comment 2: Figure 3C. There is an obvious inconsistency between the term ‘weakly bound state’ and the image, where myosin heads are detached from actin. The term 'weakly bound' refers to a biochemical state; physically, these heads are attached to actin but incapable of making a power stroke (see e.g. Ferenczi et al., Structure, 13.1, doi: 10.1016/j.str.2004.11.007). So the figure has to be corrected.

Response: We appreciate this valuable feedback. The previous Figure 3C has been renumbered as Figure 4C, and the illustration has been revised to more accurately depict the weakly bound state, in accordance with the reviewer’s comment.

Comment 3: The term ‘ATPase’ (lines 362 and 372), as it is used there, is misleading, making an impression that this is an additional separate participant of the process, whereas ATPase is myosin itself.

Response: In accordance with the reviewer’s suggestion, we have revised the sentence to read: “(B) Adenosine triphosphate (ATP) bound to the myosin head is hydrolyzed by the intrinsic ATPase activity of myosin into adenosine diphosphate (ADP) and inorganic phosphate (Pi).” (lines 384–386).

Comment 4: Fig. 4, the right panel. The biochemical SRX state, as correctly mentioned in the text, is structurally kept in the IHM, i.e., the two heads interact with each other (e.g., in Schmid and Toepfer in the refs. list or Craig and Padrón, J Gen Physiol.,2022, doi:10.1085/jgp.202113012). In the drawing, however, they are separated.

Response: We appreciate this valuable feedback. The previous Figure 4 has been renumbered as Figure 5, and the illustration has been revised so that the two heads interact with each other, in accordance with the reviewer’s comment.

Comment 5: Lines 208, 209: 100 μm, not 100 mm.

Response:  In accordance with the reviewer’s suggestion, we have corrected the unit to “100 μm” (lines 228–229).

Comment 6: Line 253: EBM? I guess it should be EMB (endomyocardial biopsy).

Response:  In accordance with the reviewer’s suggestion, we have corrected the abbreviation to “EMB” (line 274).

Comment 7: Line 354: …troponin I inhibits the actin–myosin interaction. Troponin I does not directly inhibit the interaction. It keeps tropomyosin bound to actin in the blocking state, thereby closing the myosin binding sites on its surface.

Response: We appreciate this valuable feedback. In accordance with the reviewer’s suggestion, we have revised the sentence to read: Troponin T anchors the complex to tropomyosin, whereas troponin I maintains tropomyosin in its blocking position on actn, thereby preventing access to the myosin binding sites on the actin surface. (lines 375–377).

Comment 8: It is advisable to decode abbreviations CYP2C19, CYP3A4, CYP2C9, and NT-proBNP.

Response: In accordance with the reviewer’s suggestion, we have decoded CYP as “cytochrome P450” (lines 504) and NT-proBNP as “N-terminal pro–brain natriuretic peptide” (line 129).

Reviewer 2 Report

Comments and Suggestions for Authors

Review of Nakamura et al., 2025. A review article.

Mechanism of Action and Role of Myosin Inhibitors in Hypertrophic Cardiomyopathy.

Kazufumi Nakamura 1,2,*, Takahiro Okumura 3,4, Seiya Kato 5, Kenji Onoue 6, Toru Kubo 7, Hidemichi Kouzu 8, 4 Toshiyuki Yano 8 and Takayuki Inomata 9

Summary of paper for the Authors
This is a review article by Nakamura et al on “Mechanism of Action and Role of Myosin Inhibitors in Hypertrophic Cardiomyopathy.” The review describes the discovery and history of HCOM, the pathophysiology and clinical phenotypes of HCM, and mechanism for muscle relaxation and contraction. It focusses on two cardiac myosin inhibitors, Mavacamten and Aficamten.

Critique of paper for the Authors
In general, more appropriate citations (references) should be included throughout the MS, especially where statements are made. Citations in the legends will also be useful. The figures need some revision as outlined below.
1. The title needs revision as it does not describe the mechanism of action of myosin inhibitors in detail. Several key papers that report on mechanistic effects of Mavacamten on cardiac myosin are not discussed. What kinetic/structural/physiological steps do Mavacamten and Aficamten affect? How does that result in hypercontractility?
In fact, there is significant emphasis on the clinical trials of the myosin inhibitors.
2. The review article has many statements that are not supported by appropriate citations. For e.g. Lines 66-67, 71-72, 86-88, 90-91.
3. In Fig 2A. the asterisk obscures the scar tissue, but an arrow would be useful for clarity.
4. If the panels in Figure 2 are not the authors’ original work, then citations in the legend for the panels in Figure 2 are needed, even though they were given in the MS text.
5. Line 336: what is AI-enhanced electrocardiography?
6. Citations for sentences in section 5.1 are needed.
7. In Fig 3C, part of TM structure is missing from the upper actin filament.
8. In lines 373-374 “The actin filament bound to the myosin head slides along the myosin filament owing to this structural change; this is called the power stroke (Figure 3B)” . This is an awkward sentence as the actin filament do not slide along the myosin filament, rather the “filaments slide past each other”. Also, Fig 3B shows the actin filament as a stationary unit, the cartoon should depict the movement of the actin filament.
9. Lines 304-305 should have a citation for Cooke et al paper that discovered the SRX state.
10. In Fig 4, the cartoon for SRX myosin is inaccurate as one head of the myosin molecule interacts with the other head and then the unit is folded back on the S2 region. The cartoon does not reflect this information. The legend should have appropriate citations for the statements. 11. In section 5.3, the authors cite two main proteins (myosin and MBPC3) that have variants that results in HCM. It should be noted that there are 10 sarcomeric proteins that cause HCM. It is misleading to just mention two proteins.
12. In section 5.4, it is recommended that the authors also include the citation by Rohde et al., 2018, PNAS 115(32):E7486-E7494. doi: 10.1073/pnas.1720342115 for more in-depth mechanistic effects of Mavacamten on myosin.
13. Original citations for key concepts should be included in the review when first mentioned such as for IHM (Wendt and Taylor 1999, 2001), SRX (Hooijman et al., 2011) etc.
14. In Fig. 5, it would be useful to include numbers for the segments that match the unanswered question in the legend.

Author Response

Response to reviewer 2

We greatly appreciate the reviewer’s insightful comments.

Comment 1. The title needs revision as it does not describe the mechanism of action of myosin inhibitors in detail. Several key papers that report on mechanistic effects of Mavacamten on cardiac myosin are not discussed. What kinetic/structural/physiological steps do Mavacamten and Aficamten affect? How does that result in hypercontractility?
In fact, there is significant emphasis on the clinical trials of the myosin inhibitors.

Response: We appreciate this valuable feedback. In accordance with the reviewer’s suggestion, we have revised the title to: “Cardiac Myosin Inhibitors in Hypertrophic Cardiomyopathy: From Sarcomere to Clinic.” In addition, we have expanded paragraph 5.4 (Mechanism of Action of Myosin Inhibitors) to provide a more detailed description of the kinetic, structural, and physiological steps affected by mavacamten and aficamten, explaining how these contribute to the reduction of hypercontractility. We have also incorporated key references (70, 71, and 72) to support these mechanistic insights.

Comment 2. The review article has many statements that are not supported by appropriate citations. For e.g. Lines 66-67, 71-72, 86-88, 90-91.

Response: In accordance with the reviewer’s suggestion, we have incorporated key references as follows: #6 (lines 66–67), #9 (lines 71–72), #2, #12, and #16 (lines 94–96; previously lines 86–88), and #2 and #12 (lines 99–100; previously lines 90–91).

Comment 3. In Fig 2A. the asterisk obscures the scar tissue, but an arrow would be useful for clarity.

Response: In accordance with the reviewer’s suggestion, we have replaced the asterisk with arrows to indicate the scar tissue in Figure 3A (previously Figure 2A).

Comment 4. If the panels in Figure 2 are not the authors’ original work, then citations in the legend for the panels in Figure 2 are needed, even though they were given in the MS text.

Response: All pathological images used in Figure 2 are our original work and are not reproduced from previous publications. This has been noted at the end of the figure caption as follows: “A–H: Original artwork for this article.” (lines 236-237).

Comment 5. Line 336: what is AI-enhanced electrocardiography?

Response: In accordance with the reviewer’s suggestion, we have revised the term to: “artificial intelligence (AI)-based analysis of digital data from standard 12-lead electrocardiography” (lines 358–359; previously line 336).

Comment 6. Citations for sentences in section 5.1 are needed.

Response: We appreciate this valuable feedback. In accordance with the reviewer’s suggestion, we have cited reference #62 (line 374).

Comment 7. In Fig 3C, part of TM structure is missing from the upper actin filament. 

Response: We appreciate this valuable feedback. In accordance with the reviewer’s suggestion, we have added the missing TM structure in Figure 4C (previously Figure 3C).

Comment 8. In lines 373-374 “The actin filament bound to the myosin head slides along the myosin filament owing to this structural change; this is called the power stroke (Figure 3B)” . This is an awkward sentence as the actin filament do not slide along the myosin filament, rather the “filaments slide past each other”. Also, Fig 3B shows the actin filament as a stationary unit, the cartoon should depict the movement of the actin filament.

Response: We appreciate this valuable feedback. In accordance with the reviewer’s suggestion, we have revised the sentence as follows: “This process causes structural changes in the myosin head, generating the power stroke and resulting in the actin and myosin filaments sliding past one another (Figure 4B) [63].” (lines 395–397; previously lines 373–374). In addition, we have revised Figure 4B (previously Figure 3B) to more accurately depict the movement of the actin filament.

Comment 9. Lines 304-305 should have a citation for Cooke et al paper that discovered the SRX state. Response: We appreciate this valuable feedback. In accordance with the reviewer’s suggestion, we have cited references #54-57 (lines 325-327; previously lines 304-305).

Comment 10. In Fig 4, the cartoon for SRX myosin is inaccurate as one head of the myosin molecule interacts with the other head and then the unit is folded back on the S2 region. The cartoon does not reflect this information. The legend should have appropriate citations for the statements.

Response: We appreciate this valuable feedback. In accordance with the reviewer’s suggestion, we have revised Figure 5 (previously Figure 4) to accurately depict the head–head and S2 interactions in SRX myosin, and we have added references #58 and #59 to the figure legend (line 419).

Comment 11. In section 5.3, the authors cite two main proteins (myosin and MBPC3) that have variants that results in HCM. It should be noted that there are 10 sarcomeric proteins that cause HCM. It is misleading to just mention two proteins.

Response: We appreciate this valuable feedback. In accordance with the reviewer’s suggestion, we have revised the sentence as follows: “HCM is primarily associated with pathogenic variants in more than 10 sarcomeric proteins, among which the β-myosin heavy chain (encoded by MYH7) and MYBPC3 (encoded by MYBPC3) are the major contributors [2, 12].” (lines 428–430).

Comment 12. In section 5.4, it is recommended that the authors also include the citation by Rohde et al., 2018, PNAS 115(32):E7486-E7494. doi: 10.1073/pnas.1720342115 for more in-depth mechanistic effects of Mavacamten on myosin.

Response: We appreciate this valuable feedback. In accordance with the reviewer’s suggestion, we have cited reference #71 in section 5.4 (lines 445 and 457).

Comment 13. Original citations for key concepts should be included in the review when first mentioned such as for IHM (Wendt and Taylor 1999, 2001), SRX (Hooijman et al., 2011) etc. Response: We appreciate this valuable feedback. In accordance with the reviewer’s suggestion, we have cited reference #66 and #67 (Wendt and Taylor 1999, 2001) at line 411 for IHM, and reference #56 (Hooijman et al., 2011) at line 410 for SRX.

Comment 14. In Fig. 5, it would be useful to include numbers for the segments that match the unanswered question in the legend.

Response: We appreciate this valuable feedback. In accordance with the reviewer’s suggestion, we have revised Figure 6 (previously Figure 5) to add numeric labels to each segment, and we have updated the figure legend accordingly.

Changes in the manuscript:

  • Figure 5 now displays segment numbers (1…n).
  • The figure legend has been rewritten to list the segment numbers (1, 2, …) and to identify the segment corresponding to the unanswered question.

Reviewer 3 Report

Comments and Suggestions for Authors

The review provides a comprehensive and well-structured overview of the treatment of hypertrophic cardiomyopathy (HCM) with myosin inhibitors, highlighting both the mechanistic insights and clinical implications of this novel therapy.  The topic is vey innovative and attractive for both research community and clinicians. The authors have successfully synthesized current evidence in a clear, accessible manner, making the article highly informative  for both clinicians and researchers. The balanced discussion of clinical trial data, potential benefits, and therapeutic challenges adds significant value, making this this review really outstanding, while the emphasis on future perspectives makes the review especially engaging. Overall, this is a well-written, timely, and highly relevant contribution to the field of cardiovascular medicine. I recommend the authors to address several minor issues in order to improve the article prior to acceptance:

  1. Please, avoid informal english such as `autosomal dominant fashion` in introduction, use formal language eg. manner instead of fashion. Check throught the whole manuscript.
  2. It would be useful to add an illustration on the types od HCM and pathophysiology
  3. Please add some info regarding the role of cardiac enzymes in the diagnosis of HCM in the subsection 2.2. Diagnosis
  4. Please, comment on the possibility of HCM treatment with ARNI such as sacubitril/valsartan for example or the possibility of HCM with some other drug class besides mentioned ones (beta blockers, calcium channel inhibitors..) in the subsection 2.3. Treatment
  5. Please, correct rifampin into rifampicine
  6. It is recomended to add info regarding the metabolism reaction pathway of mavacamten, besides CYP enzyme, it can be added also as a chemical reaction figure (eg. glucuronidation), and also, add examples of drugs interacting with with mavacamten such as abatacept, benzodiazepins, abirateron..etc. or food, if there is data available

Author Response

Response to reviewer 3

We greatly appreciate the reviewer’s insightful comments.

Comment 1: Please, avoid informal english such as `autosomal dominant fashion` in introduction, use formal language eg. manner instead of fashion. Check throught the whole manuscript.

Response: We appreciate the reviewer’s valuable feedback. In accordance with the suggestion, we have revised the term “fashion” to “manner” in the Introduction (line 48).

Comment 2: It would be useful to add an illustration on the types of HCM and pathophysiology

Response: We sincerely appreciate the reviewer’s valuable suggestion. Accordingly, we have included Figure 1. Subtypes of HCM in Section 2 (line 78) and referenced it again in Section 3 (line 207).

Comment 3: Please add some info regarding the role of cardiac enzymes in the diagnosis of HCM in the subsection 2.2. Diagnosis

Response: We appreciate this valuable feedback. In accordance with the reviewer’s suggestion, we have added the following sentences: “Cardiac enzymes such as creatine kinase (CK) or lactate dehydrogenase (LDH) may show mild or transient elevation in some patients with HCM due to ongoing myocyte injury or microvascular ischemia; however, they are not specific for the disease and are therefore not routinely used for its diagnosis. In contrast, circulating natriuretic peptides (brain natriuretic peptide [BNP] or N-terminal pro-brain natriuretic peptide [NT-proBNP]) are often persistently elevated and more closely reflect diastolic dysfunction and hemodynamic burden in HCM, thus providing adjunctive value in disease assessment rather than initial diagnosis.” (lines 125-132).

Comment 4: Please, comment on the possibility of HCM treatment with ARNI such as sacubitril/valsartan for example or the possibility of HCM with some other drug class besides mentioned ones (beta blockers, calcium channel inhibitors..) in the subsection 2.3. Treatment

Response: We appreciate this valuable feedback. In accordance with the reviewer’s suggestion, we have added the following sentence: “In patients with obstructive physiology, vasodilators such as angiotensin-converting enzyme (ACE) inhibitors, angiotensin receptor blockers (ARBs), or angiotensin receptor–neprilysin inhibitors (ARNI, e.g., sacubitril/valsartan) may exacerbate LVOTO and are therefore generally avoided [12].” (lines 166-169).

Comment 5: Please, correct rifampin into rifampicine.

Response: In accordance with the reviewer’s suggestion, we have revised “rifampin” to “rifampicine” (line 770).

Comment 6: It is recomended to add info regarding the metabolism reaction pathway of mavacamten, besides CYP enzyme, it can be added also as a chemical reaction figure (eg. glucuronidation), and also, add examples of drugs interacting with with mavacamten such as abatacept, benzodiazepins, abirateron..etc. or food, if there is data available.

Response: We appreciate this helpful suggestion. Beyond CYP-mediated oxidation, non-CYP conjugation pathways (e.g., glucuronidation) have not been characterized in humans for mavacamten. To avoid over-interpretation, we did not include a speculative reaction-scheme figure in the present manuscript.

At the same time, we have revised the text to include concise, representative examples of clinically relevant drug–drug interactions and a brief statement on food effects. Specifically, we now cite examples of:

  • Strong CYP2C19 inhibitors (e.g., fluvoxamine, fluconazole) (lines 768–769)
  • Strong enzyme inducers of CYP2C19/3A4 (e.g., rifampin, carbamazepine, phenytoin, St John’s wort) (lines 770–771)
  • Mild-to-moderate modulators (e.g., omeprazole, diltiazem/verapamil) (lines 772–773)

In addition, we have added the following sentences: “Food has no clinically meaningful effect on exposure, and mavacamten may be administered with or without meals.” (lines 774–775) and “For a comprehensive list of interacting agents and management recommendations, clinicians should consult the most recent full prescribing information before initiation and whenever starting or stopping concomitant drugs.”(lines 781–784).
